# Multimodal Interaction between a Mother and Her Twin Preterm Infants (Male and Female) in Maternal Speech and Humming during Kangaroo Care: A Microanalytical Case Study

**DOI:** 10.3390/children8090754

**Published:** 2021-08-30

**Authors:** Eduarda Carvalho, Raul Rincon, João Justo, Helena Rodrigues

**Affiliations:** 1CESEM-NOVA-FCSH, 1069-061 Lisbon, Portugal; raulenrique.rincon@gmail.com (R.R.); helenarodrigues@fcsh.unl.pt (H.R.); 2Faculty of Psychology, University of Lisbon, 1649-004 Lisbon, Portugal; jjusto@psicologia.ulisboa.pt

**Keywords:** mother-twin preterm dyad, infant’s gender, infant gaze, infant vocalizations, maternal speech, maternal humming, affectionate maternal touch, vocal modulation, melodic contours, speech contents

## Abstract

The literature reports the benefits of multimodal interaction with the maternal voice for preterm dyads in kangaroo care. Little is known about multimodal interaction and vocal modulation between preterm mother–twin dyads. This study aims to deepen the knowledge about multimodal interaction (maternal touch, mother’s and infants’ vocalizations and infants’ gaze) between a mother and her twin preterm infants (twin 1 [female] and twin 2 [male]) during speech and humming in kangaroo care. A microanalytical case study was carried out using ELAN, *PRAAT*, and MAXQDA software (Version R20.4.0). Descriptive and comparative analysis was performed using SPSS software (Version V27). We observed: (1) significantly longer humming phrases to twin 2 than to twin 1 (*p* = 0.002), (2) significantly longer instances of maternal touch in humming than in speech to twin 1 (*p* = 0.000), (3) a significant increase in the pitch of maternal speech after twin 2 gazed (*p* = 0.002), and (4) a significant increase of pitch in humming after twin 1 vocalized (*p* = 0.026). This exploratory study contributes to questioning the role of maternal touch during humming in kangaroo care, as well as the mediating role of the infant’s gender and visual and vocal behavior in the tonal change of humming or speech.

## 1. Background

Mothers around the world communicate in a musical way with their infant by humming, singing, and talking to them. These vocal behaviors are composed of musical and acoustic elements (pulse, melodic contours, pitch) and play a crucial role in the development of communicative musicality from the neonatal period onward [1,2]. Mother–infant interactions are coordinated in a multimodal way, coexisting multiple sensory modes (visual, vocal, gestural) that are also coordinated in multiple temporal forms in the interaction. [3]. However, multimodal interaction has rarely been studied between mothers and infants less than 3 months old. Also, there is little knowledge about the characteristics of multimodal interaction between mothers and their preterm infants.

Having twins can be a condition of vulnerability for mothers as it can reduce their capacity for self-regulation and affect the quality of mother–infant interaction [4]. Having premature twins can make this interaction even more difficult.

The literature reports that maternal speech, singing, or humming directed to preterm infants in the Neonatal Intensive Care Unit (NICU) has a positive effect on infant self-regulation [5,6] and on the neurodevelopment and brain plasticity of preterm infants [7,8]. More self-exploration with the hands of the face, mouth, nose, and eyes has been found when mothers talk or sing to their preterm infants in the NICU [9]. Similar self-exploration movements of the fetus are already present at the end of pregnancy, pointing to fetal neurological maturation [10].

Although maternal vocalizations may have the effect of eliciting infant vocalizations, specific forms of contingent maternal vocal stimulation can reinforce an infant’s early vocalizations effectively [11]. Motherese is a particularly affectionate way in which mothers talk to their infants. When using motherese, mothers modulate their voices, changing their temporal and tonal parameters contingently according to the context and the signals emitted by the infants [12,13]. Infant-directed (ID) speech, when compared with non-ID speech, is characterized by a high pitch and high tonal variability and a slower tempo which attracts the infant’s attention [14]. When infants exhibit a behavioral change, maternal speech has a greater range of tonal qualities than when infants are passive and expressionless. Specifically, there is an association between the infant’s behavior and higher values of average pitch and a higher maximum sound pressure level, as well as a higher variability of these parameters [14]. The interactive context seems to influence the tonal quality (melodic contours) of speech and singing. In the context of play, melodic contours are similar in singing and speech [14,15], while in soothing contexts, more sinusoidal contours are found in singing than in speech [15].

The literature highlights the role of the behavioral state of preterm infants in the vocal modulation of the parental voice [16,17]. An increase in the pitch of the maternal voice (speech or singing) directed to preterm infants in the NICU was found after positive expressions by the infant such as opening the eyes, gazing at the mother or smiling [16]. Also, the vocal spectrum of the parents’ speech was wider when the preterm infants were sleeping, or moving from one state to another, compared to when the infants were awake [17].

The temporal organization of the face-to-face interaction between the mother and the infant plays an important role in the synchronization of the dyad [18]. This synchrony and the interconnection of vocalizations and body movements and eye contact are referenced in the literature [19,20]. Studies of early human communication emphasize the importance of several modalities of communication such as the gaze, facial expressions, body movements, touch, and vocalizations [21,22]. Despite the emphasis placed on the multimodal characteristics of the mother–infant interaction, there is little knowledge about the characteristics of co-modality among the various modalities of communication.

Infant responsiveness during speech and singing has been mostly studied from the second semester of life onward [23,24]. From 6 months of age, infants seem to give more sustained attention to extended episodes of maternal singing (shown through longer periods of visual contact and less movement) than to comparable episodes of maternal speech [23]. The repetitive patterns and low tonal variability of ID singing probably foster more moderate arousal levels, which facilitate longer episodes of infant engagement [23]. The variable tonal contours of ID speech play a key role in communicative intentionality and in understanding a message, while the melodic contours of ID singing play a crucial role in the infant’s self-regulation [15]. The rate of the infant’s vocalizations is higher in ID parental speech while the rate of the infant’s overlapping vocalizations is higher in ID parental singing [24]. More preterm infants’ overlapping vocalizations have been found during maternal humming phrases than in humming pauses while more infants’ vocalizations have been found in speech pauses than in speech phrases [25]. This suggests that humming or singing may be more favorable conditions for the vocal synchronization and attunement of dyads, while speech favors the experience of turn-taking. In addition, more overlapping vocalizations have been found during maternal humming (especially at the end of musical phrases) in preterm female infants than in preterm male infants [26].

Early interventions promoting holding such as kangaroo care and the mother’s affectionate touch lead to positive physiological and psychological outcomes such as regulating stress reactivity and reinforcing mother–infant bonding [27,28]. During face-to-face interactions, the parental use of affectionate stroking decreases the infant’s arousal and increases positive emotionality [28,29]. An affectionate touch relates inversely to the infant’s activity level while a stimulating touch correlates positively with the infant’s activity [30].

Maternal affectionate touch seems to play a role in infant self-regulation [31] and in establishing and maintaining parent–infant interactive synchrony at the behavioral and neural levels [32]. The role played by affectionate maternal touch and speech in the context of invasive procedures for preterm infants in the NICU has been highlighted in the literature [33,34]. However, there has been little research into the interconnection and synchronization of maternal touch and maternal voice (speaking and singing) addressed to the preterm infant.

The literature underlines a higher synchrony within mother–daughter dyads compared with mother–son dyads [35]. This suggests that synchronization can be seen as phylogenetically determined as the result of gender difference [35]. Moreover, male infants take more time to achieve self-regulation and need more time to move from a unilateral to a mutually engaged state than female infants [36] and at 5–12 months old, male infants were more likely than female infants to cry or whine when their mothers disciplined them [37]. A longitudinal study found that mothers of male infants touch their sons more frequently than mothers of female infants touch their daughters at 3–4 months of age [38]. Mothers give different messages to their male and female infants [39,40]. Mothers of female infants make more interpretations and engage in more conversations whereas mothers of male infants make more comments, and their attention is characterized by instructions rather than conversations [41]. Furthermore, mothers use longer vocalizations, more words, and more affectionate terms with their sons than with their daughters [37]. However, the role of the infant’s gender in the quality of the multimodal interaction of preterm dyads during maternal speech and singing or humming is still poorly understood.

The importance of multimodal interaction in preterm dyads has been emphasized in the practice of music therapy in neonatal care through the promotion of the use of the contingent maternal humming particularly during kangaroo care [42]. However, we know little about the specificity of how the maternal touch (during the kangaroo method) and maternal vocalizations (in the modalities of humming and speaking) coordinate with each other.

The aim of the present study is to deepen knowledge about the temporal organization of multimodal interaction between a mother and her twin preterm infants, one female (twin 1) and one male (twin 2) regarding the maternal vocalizations, the maternal touch, the infant’s vocalizations and the infant’s gaze during the humming or speech conditions.

Specifically, we intend to know the temporal characteristics (frequency and length) concerning: (1) maternal vocalizations (or phrase units) in humming and in speech conditions; (2) the maternal touch units that overlapped on the vocalizations of humming or speech, (3) the infant’s vocalizations in each of the conditions, (4) the infant’s gaze episodes in each of the conditions; we also intend to know: (5) if the occurrence of any of the infant’s behaviors (vocalizations or gaze) can affect the maternal pitch of speech or humming, (6) the proportions of the melodic contours of maternal humming (linear, rising, falling, bell-shaped, U-shaped, and sinusoidal) directed at twin 1 and twin 2 and, finally, (7) the proportion of verbal contents of speech directed to twin 1 and twin 2.

## 2. Method

### 2.1. Sample

The sample for this case study was recruited from a previous study [25] with a larger sample of 36 preterm mother–infant dyads in the NICU of the Doutor Alfredo da Costa Maternity Hospital in Lisbon, Portugal. The case study was of a mother and her twin infants, one female (twin 1) and one male (twin 2).

The mother, an Angolan national from Luanda, was 42 years old. She had a high educational level with 21 years of education. She was a medical doctor specializing in ophthalmology. She was living with her parents and at the time of the interview she had no marital partner and no other children. She reported having had two pregnancies that ended in spontaneous abortions when she was 30 and 41 years old respectively; for the second of these pregnancies, artificial insemination with donor sperm was used for the first time. The subsequent twin pregnancy used artificial insemination with gametes provided by an unknown donor. Preterm delivery occurred as the result of maternal sickle cell anemia and the restricted growth of both fetuses. The maternal perception of body movements of the male fetus occurred around 20 weeks of gestation, while the body movements of the female fetus were perceived at 24 weeks of gestation.

The preterm twins, one female (twin 1) and one male (twin 2), were born after a cesarean section at 32 weeks of gestation. Twin 1 was born first. At birth, twin 1 had a weight of 1340 g, length of 48 cm, Apgar score at the first minute of 9 and at the fifth minute of 10. At birth, twin 2 had a weight of 1300 g, length of 47 cm, Apgar score at the first minute of 8 and at the fifth minute of 9. Twin 2 was observed at 34 weeks of gestation, one week before twin 1 who was observed at 35 weeks of gestation. At the time of observation twin 2 weighed 1525 g and twin 1 weighed 1600 g.

Both infants had only one kangaroo care experience prior to the observation. During the first kangaroo care experience, before observation, the mother reported that her son reacted very well to kangaroo care; he was attentive at first with his eyes open calmly and he ended up falling asleep. Her daughter, in contrast, was very agitated; she was crying and took time to adapt.

The mother’s general perception of twin 2 was of him being calm and attentive, opening his eyes frequently when the mother talked to him and reacting positively to her touch and speech. The mother described twin 1 as being more agitated and less tolerant of maternal touch but also as loving being held on her mother’s lap. The mother reported several ways of interacting with her infants, including looking, touching, talking, singing, caressing, and using music.

### 2.2. Design

The data collection for this case study was partially carried out in a previous microanalytical study with a larger sample of preterm dyads (N = 36) including mothers with single or twin infants. The aim of the previous study was to show the temporal features of maternal speech and humming directed to a preterm infant in kangaroo care, placing the infant in a diagonal position—known as kangaroo supported diagonal flexion positioning—favoring visual contact [43].

Another aim was to understand the infant’s vocal responsiveness during these two vocal conditions [25]. The previous study was approved by the Ethics Committee of Central Lisbon Hospital Center (267/2015). After signing an informed consent form, mothers were interviewed about sociodemographic and clinical characteristics.

A video-recorded observation (using a camera Panasonic 4K HC-VX870-MP4) was performed for each dyad. For the observation of the mother–infant interaction dyad each mother was asked to interact with the infant during kangaroo care, talking and humming to them according to the following instruction: “This observation will consist of a sequence of 5 periods (each lasting 3 min) in which I give you a signal to change your condition of: (1) remain silent, (2) talk to the infant (as usual do), (3) go back to silence, (4) hum to the infant (using an improvised melody with a closed mouth without using words) and finish in silence again”. The sequence was counterbalanced to control the order effect. The participants with odd numbers were assigned the sequence where the speaking condition occurred before the humming condition and the participants with even numbers were assigned the sequence where the humming condition occurred before the speaking condition. The mother in the present case study performed the same sequence for twin 1 and twin 2 (with speech as the first vocal condition and humming as the second vocal condition).

After the interaction had been observed, each mother was interviewed about her experience during the vocal interaction, including her perception of the infant’s behavior when she spoke and hummed to the infant.

### 2.3. Microanalytic Coding

Several microanalytic codings were carried out using: (1) ELAN software (EUDICO Linguistic Annotator, version 4.9.4) to code the frequency (number) and the length of maternal and infant vocalizations, maternal affectionate touch (touch units), the infant’s gaze and the melodic contours of humming, (2) *PRAAT* software (Version V6.0.29) to code the tonal features (F0, F1) of speech and humming and if and how the infant behavior (gaze or vocalizations) of twin 1 and twin 2 relate to the tonal quality (F0, F1) of speech and of humming, and (3) MAXQDA software to code the verbal content of maternal speech.

The temporal criteria established by Gratier and her team [44] were used to code each maternal vocalization (defined as a phrase of speech or humming). According to these criteria, a maternal vocalization (speech or humming phrase) was coded when the interruption of an audible (and visualized in the spectrogram) vocal segment produced by the mother did not exceed 300 ms. A touch unit was identified when the mother’s hand caressed the face or hands of the infant when speaking or humming to him. A touch unit was coded from the moment the mother’s hand or fingers touched the infant’s face or hands moved across her skin until the touch was stopped. A vocalization of the infant was coded when an audible vocal sound of the infant such as murmurs, sighs, or whining was visualized in the spectrogram. A gaze episode of the infant was codified from the moment the infant opened his eyes (towards the mother’s face).

Falk’s protocol [15] was used to place the melodic contours of humming into six categories: (1) linear (when the musical tone is flat), (2) rising (when the musical tones ascend), (3) falling (when the musical tones descend), (4) bell-shaped (a sequence of ascending then descending musical tones), (5) U-shaped (a sequence of musical tones descending and then ascending), and (6) sinusoidal (any sequence of U-shaped and bell-shaped contours).

Maternal speech content directed to twin 1 and to twin 2 was categorized using the following categories and subcategories: (1) needs (speech aimed at the satisfaction of the infant’s needs or the satisfaction of maternal needs), (2) infant designation (first name, diminutive, and affectionate words), (3) physical features of the infant (positive and negative physical features), (4) temperamental features of the infant (positive and negative temperamental features), (5) maternal worries (infant’s sleep, breastfeeding/weight, comfort/temperature and holding the infant on the lap), and (6) maternal desires (infant development, hospital discharge, and family union).

In order to verify whether the behaviors of each of the infants had an effect upon increasing or decreasing the pitch and the intensity of maternal speech we selected the maternal vocalizations (phrases) that preceded and followed any vocalization or a gaze episode of twin 1 or twin 2. *PRAAT* software was used to make a comparison between the acoustic characteristics (mean pitch and intensity) of maternal speech, before and after any of the visual or vocal behaviors of each of the twin infants. Likewise, to find out if the infants’ behaviors had an effect on increasing or decreasing the pitch and intensity of maternal humming, we selected the musical segments that preceded and followed the infants’ vocal or visual behaviors. The audio segments were analyzed in *PRAAT* to identify acoustic variables related to pitch. Each of these humming segments was acoustically compared to humming segments that, from the melodic point of view, were similar but not preceded or followed by any of the infants’ behaviors.

Finally, several descriptive and comparative statistical analyses of the microanalysis data were performed using SPSS Statistics 25 software.

### 2.4. Reliability

Two researchers performed independent coding and the inter-rater reliability was calculated through intra-class correlation coefficients (ICC). Regarding the frequency (number) of the total maternal vocalizations (phrases), there was a high rate of agreement for the frequency (number) of the speech phrases, (ICC = 0.991, *p* = 0.000, df = 136) as well as for the humming phrases (ICC = 0.999, *p* = 0.001, df = 133). A high rate of agreement was found for the frequency (number) of total touch units during the speech (ICC = 0.895, *p* = 0.002, df = 50) as well as during the humming (ICC = 0.899, *p* = 0.001, df = 43).

In terms of the frequency (number) of total infants’ vocalizations, coefficients were also high particularly for the frequency of vocalizations during speech (ICC = 0.985, *p* = 0.000, df = 9) as well as for the frequency of vocalizations in humming (ICC = 0.996, *p* = 0.000, df = 28). The rate of agreement for the frequency of total infant episodes gaze in speech was high (ICC = 0.889, *p* = 0.001, df = 15) and for the frequency of total infant episodes gaze in humming (ICC = 0.887. *p* = 0.001, df = 5).

Regarding the frequency (number) of total melodic contours of humming, an additional reliability analysis was performed by two researchers with a musical background achieving a high degree of agreement in all melodic contours identified: rising (ICC = 0.999, *p* = 0.000, df = 27), bell-shaped (ICC = 0.995, *p* = 0.000, df = 25), U-shaped (ICC = 0.933, *p* = 0.002, df = 17) and sinusoidal (ICC = 992, *p* = 0.000, df = 64).

The frequency (number) of the total categories/subcategories of speech contents that had agreement coefficients were the following: (1) infant’s needs (ICC = 0.862, *p* = 0.000, df = 38), (2) maternal needs (ICC = 898, *p* = 0.000, df = 32), 3) first name of the infant (ICC = 1, df = 17), (4) diminutive of the first name (ICC = 1, df = 7), (5) affectionate words (ICC = 1, df = 21), (6) infant’s physical features (ICC = 0.990, *p* = 0.000, df = 2), (7) infant’s temperamental features (ICC = 0.1, df = 1), (8) maternal worries (ICC = 0.898, *p* = 0.000, df = 29), and maternal desires (ICC = 0.980, *p* = 000, df = 4).

## 3. Results

### 3.1. Maternal Affectionate Touch and Maternal Vocalizations (Phrases) in Speech and in Humming

Considering the first and second aims of this study (to analyze maternal vocalizations and maternal touch in speech and in humming), descriptive and comparative statistical analyses were performed. Table 1 shows the results of these statistical descriptive analyses. Regarding the maternal vocalizations directed to both infants, humming phrases (twin 1 M = 1880.86 ms twin 2, M = 2355.50 ms) were longer than speech phrases (twin 1, M = 1514.09 ms twin 2, M = 1438.14 ms) and differences are statistically significant for the twin 1 (t = −2.531; df = 71; *p* = 0.014) and for the twin 2 (t = −8.149; df = 60; *p* = 0.000). However, there were similar numbers of speech phrases and humming phrases for twin 1 and for twin 2.

Comparing the lengths of humming vocalizations (humming phrases) directed to the twin 1 (M = 1880.86) and to the twin 2 (M = 2355.50), a significant difference was found (t = −3.098; df = 120.198; *p* = 0.002; Bonferroni correction, *p* = 0.025). However, no significant differences (t = 0.502; df = 133.928; *p* = 0.615) were found in the lengths of speech vocalizations (speech phrases) directed to the twin 1 (M = 1514.09 ms) and to the twin 2 (M = 1438.14 ms).

Regarding the touch units, we noticed that when the mother was talking or humming to her daughter (twin 1) she caressed her daughter’s face. Meanwhile, when the mother was with her son (twin 2), she caressed his hand and only when she spoke to him and not when she hummed to him. Table 1 displays the frequencies and lengths of the overlapping touch units in speech and in humming addressed to twin 1 and to twin 2. For the mother–twin 1 dyad, there were a similar number of temporal segments of touch in speech (*n* = 47) and in humming (*n* = 43) but the average length of these segments in humming (M = 2174, 65 ms) was longer than in speech (M = 1252, 87 ms) and the difference was significant (t = −5.454; df = 42; *p* = 0.000). For the mother–twin 2 dyad, only three temporal segments of touch were found during speech, and none was observed in humming.

Proportions of the touch units that overlapped maternal vocalizations (phrases) in speech and in humming were calculated in relation to the total number of maternal vocalizations (phrases) in speech and in humming addressed to each infant. In the mother-twin 1 dyad, the temporal units of overlapping touch occurred more often in speech (65.27%) than in humming (59.72%), although they were longer in humming (M = 2174.65 ms) than in speech (M = 1252.87 ms). There was also a higher proportion of touch units during speech vocalizations (phrases) directed to twin 1 (65, 27%) than to twin 2 (4, 68%). However, touch units that overlapped with speech vocalizations (phrases) were longer when directed to twin 2 (M = 1310.00 ms) compared to twin 1 (M = 1252.87 ms) and the difference was significant (t = 7.744; df = 92.162; *p* = 000; Bonferroni correction, *p* = 0.025).

### 3.2. Gaze Episodes and Vocalizations of Twin 1 and Twin 2 in Speech and Humming

The third and fourth aims of this study are related to the analysis of infant vocalizations and the gaze of twin 1 and twin 2 during maternal speech and humming. Several descriptive statistical and comparative analyses were carried out. Table 2 shows the results of the descriptive analysis relative to the frequency (number) and length of gaze episodes and vocalizations of twin 1 and twin 2 in speech and in humming.

The infant vocalizations of twin 1 were more frequent and longer in humming (*n* = 28; M = 263.75 ms) than in speech (*n* = 9; M = 245.55 ms) but the difference between the average lengths is not statistically significant (t = −0.354; df = 8; *p* = 0.732). The gaze episodes of twin 1 were only observed in speech. For twin 2, we found a similar number of gaze episodes in speech (*n* = 4) and in humming (*n* = 5). The gaze episodes of twin 2 were longer on average in speech (M = 2737.50 ms) than in humming (M = 1726.00 ms) but this difference is not statistically significant (t = 1.200; df = 3; *p* = 0.316). Twin 2 made no vocalizations in speech or in humming. Although in speech twin 1 had more gaze episodes (*n* = 11) than twin 2 (*n* = 4), the gaze episodes of twin 2 were longer (M = 2737.50 ms) than the gaze episodes of twin 1 (M = 1705.45 ms). However, this difference is not statistically significant (t = 0.710; df = 129.911; *p* = 0.479).

### 3.3. Infant Behavior (Vocalizations and Gaze Episodes) and the Tonal Quality of Maternal Speech and Humming

The fifth aim of this study is to understand the relationship between the vocalizations of both twin infants and the changes that occur in the pitch (Hz) of maternal speech and humming. Also, we intend to understand the relationship between the gaze episodes of both twin infants and maternal pitch variations in speech and humming. For this, we first present a descriptive analysis and then a comparative analysis. Table 3 shows the results of the descriptive statistical analysis of the pitch of maternal speech and humming before and after the gaze episodes of twin 1 and twin 2, and before and after the vocalizations of twin 1 and twin 2 in both observed conditions.

A comparative analysis between the pitch (Hz) of maternal humming before (M = 249.28 Hz) and after (M = 255.06 Hz) a vocalization by twin 1 showed a statistically significant increase (t = −4.099; df = 3; *p* = 0.026) after the infant’s vocalization. In contrast, a decrease in the mean pitch of maternal speech to twin 1 was found after the infant’s vocalization (before: M = 250.96 Hz; after: M = 224.66 Hz), but this decrease is not statistically significant (t = 1.640; df = 2; *p* = 0.243). In addition, we observed an increase in the pitch of maternal speech directed to twin 1 (before: M = 223.01 Hz; after, M = 230.31 Hz) and to twin 2 (before: M = 209.87 Hz; after: M = 233.59 Hz) after an infant gaze episode. While the increase for twin 1 is not significant (t = −0.639; df = 5; *p* = 0.551) it is significant for twin 2 (t = −9.822; df = 3; *p* = 0.002). In contrast, a decrease in the pitch of maternal humming to twin 2 (before: M = 201.88 Hz; after: M = 156.44 Hz) was observed after an infant gaze episode but this is not significant (t = 0.751, df = 3; *p* = 0.507).

### 3.4. Melodic Contours of Humming Directed to Twin 1 and to Twin 2

In order to analyze the melodic contours of the humming directed to both twin infants (the sixth aim of this study), ELAN software (Version 11.3.8.2) was used. Table 4 presents the results of the descriptive statistical analysis of the number and length of each melodic contour type (linear, rising, falling, bell-shaped, U-shaped, and sinusoidal). The proportions of each melodic contour were also calculated relative to the total number of maternal humming vocalizations (phrases) directed to twin 1 (*n* = 72) and to twin 2 (*n* = 61). For maternal humming directed to twin 1, we identified a higher proportion of rising contours (37.5%) and sinusoidal contours (34.72%), followed by bell-shaped contours (20.83%) and U-shaped contours (8.33%). No falling or linear contours were identified. For the maternal humming directed to twin 2, we found a higher proportion of sinusoidal contours (63.93%) followed by U-shaped contours (18.03%), bell-shaped contours (16.39%). No falling, rising, or linear contours were identified. The proportion of sinusoidal segments found in humming to twin 2 was remarkably higher than in humming to twin 1.

To complement the analysis of melodic contours we used an oscillogram (using *PRAAT* software) to analyze the tempo in beats per minute (bpm) of the humming directed to each infant. According to the results, the average bpm for humming directed to twin 2 was slower (M = 81.38 bpm, min. = 77.08, max. = 85.09) than the mean value of the bpm of the humming directed to twin 1 (M = 91.07 bpm, min. = 86.44, max. = 101.3). Furthermore, we found that the maximum bpm value of the humming directed to twin 1 occurred as a result of the tonal synchrony between the infant’s overlapping vocalizations and the musical notes of the maternal humming.

### 3.5. Speech Contents Directed to Twin 1 and to Twin 2

In order to analyze the speech content (the final aim of this study) MAXQDA software was used. Table 5 shows examples of speech segments directed to twin 1 and twin 2 that were codified using categories and subcategories.

The proportion of vocalizations in each content category and subcategory was identified relative to the total number of speech maternal vocalizations. Table 6 shows the frequencies and the proportions (according to the total maternal speech vocalizations) of the speech contents directed to the female and to the male infant.

According to the results, maternal speech directed to twin 2 seemed to be more oriented toward satisfying the son’s needs than toward satisfying the mother’s needs while maternal speech directed to twin 1 seemed to be oriented simultaneously toward the satisfaction of the daughter’s needs and the mother’s needs. Although the mother addressed both infants using affectionate words, she made more reference to the first name of her daughter (twin 1) than her son (twin 2) and diminutives (of the first name) were only found in speech addressed to her daughter. In addition, the physical and temperamental features of the infant were only referred to when the mother spoke to her daughter. The main maternal concern was about the infant’s sleep which was observed for both infants, but particularly for twin 2. The maternal desire for the infant’s development was expressed only for twin 1, while the desire for hospital discharge and family union was only present for twin 2.

## 4. Discussion

The present study expands the knowledge of the characteristics of a multimodal interaction between a mother and her twin preterm infants when the mother speaks and hums to her infants during kangaroo care using diagonal positioning to facilitate eye contact. According to the results we found: (1) for both infants significantly longer humming phrases than speech phrases, but significantly longer humming phrases for twin 2 than for twin 1, (2) a longer touch units in humming than in speech to twin 1, (3) no statistically significant differences regarding infant’ vocalizations and infants’ gaze episodes in both humming and speech conditions, (4) an increase in humming pitch after twin 1 had vocalized and an increase in speech pitch after twin 2 had opened his eyes towards the maternal face, (5) only rising contours in humming for twin 1 and a higher proportion of sinusoidal contours in humming directed to twin 2, and (6) verbal contents of the speech directed to twin 1 that seem to be associated with a representation of greater autonomy, while the contents of the speech directed to twin 2 that seem to be associated with a greater protection.

In addition to the quieter behavior and younger age of twin 2, the pattern of findings in this case study suggests that the mother may have perceived twin 2 as more vulnerable. We might think that this perception may have triggered special care in maintaining the infant’s self-regulation, using for this a soft humming style characterized by longer humming phrases with a higher proportion of sinusoidal contours (similar to a lullaby). In contrast, the more vocal responsiveness and greater maturity of twin 1 (35 weeks of gestation) may have prompted the mother to perceive the infant as being more able to interact, inducing the use of a more exciting song composed of shorter phrases with a sinusoidal and rising contour (similar to a play song). Likewise, the bpm of the humming directed to twin 2 was lower than the bpm of the humming directed to the twin 1, emphasizing once again the mother’s intention to hum for her son in order to lessen his arousal state while she hummed to her daughter in order to encourage her participation. This suggests that maternal humming can be influenced by the infant’s behavior and is contingently adjusted to the infant’s needs and infant’s signals. This underlines the crucial role of maternal humming directed to preterm infants in the NICU [5,6,9], particularly to improve the infant’s brain plasticity and neurodevelopment [7,8].

Regarding maternal touch, when the mother addressed her daughter there was more co-modality between touch and humming than when she addressed her son. The literature emphasizes a greater synchrony in mother–daughter dyads than in mother–son dyads, suggesting that the synchronization seems to be phylogenetically determined [35]. As indicated by the literature [36], the mother–daughter interaction is guided early on in both a multimodal and a mutual way, facilitating the synchrony of the dyad, while the mother–son interaction seems to be organized in a unimodal and unilateral way. However, more studies are needed to deepen understanding of such gender differences. We might ask whether the maternal touch during humming favors synchronization and attunement, particularly in the mother–daughter dyads. Interestingly, the co-modality between maternal touch and vocalizations directed to twin 1 was sustained for longer periods in humming than in speech. The longer periods of maternal touch when the mother hummed to twin 1 could have been a strategy to keep twin 1 in a moderate state of arousal and self-regulation, since the mother described her daughter as being more restless and less tolerant of maternal touch than her son. According to the literature, affectionate maternal touch seems to play a role in positive emotionality [28,29] and in infant self-regulation [30,31]. Perhaps the repeated patterns of rhythm, melodic contours, and tempo that characterize humming [23] favor the use of affectionate maternal touch in a more sustained and synchronized. This underlines the role of affectionate maternal touch and humming in promoting the attunement of the preterm dyad and in facilitating attachment in the NICU.

Our study questions the impact of infant behavior (gaze and vocalizations) upon maternal vocal modulation (during speech and humming) and asks whether this is mediated by the infant’s gender or by the maternal expectations of the infant. Also, a previous study in the context of the NICU found an increase in the pitch of maternal speech directed to preterm infants after the infants made positive expressions like opening their eyes, looking at their mothers or smiling [16]. In addition to contributions about gender and the different visual or vocal behaviors of infants, there may be individual differences between the twin 1 and the twin 2 that can also contribute to interaction. We may question whether the mother perceived twin 2′s gaze as an attention response, thus increasing the pitch to reinforce the infant’s attention. Also, we can ask if the increase in tone (pitch) of the maternal humming after the vocalization of twin 1 could have been could it have been a strategy to maintain the infant’s vocal responsiveness.

Similarly, when the mother hummed to her son (twin 2) using a greater proportion of calming sinusoidal contours (in a lullaby style) she seems to have directed her speech to the infant to ensure his self-regulation by guiding the speech towards the satisfaction of her son’s needs (“How are you there? You’re comfortable, are you?”) more than her own needs. However, when the mother talked to twin 1, she seemed to meet her daughter’s needs (“Hi! How was your night?”) as well as her own needs (“Mother missed you so much”). When the mother was speaking to the infants she used affectionate words for both infants (to twin 1: “So my little princess?” and to twin 2: “Hello my Prince! Mom’s youngest!”) but when the mother addressed her daughter, she used her first name more (“Where is Beatriz?”) than when she addressed her son (“Hi David!”). Only speech directed to twin 1 referred to the infant’s physical features (“You’re opening your little eye, are you?”) and the infant’s temperamental features (“Very tired? Poor girl!”) suggesting a differentiated representation of the daughter. The concern about the infant’s sleep is present in speech to both infants, but particularly to twin 2 (for twin 1: “Did you sleep well tonight?” and for twin 2: “Hmm! how sleepy I am!”). Regarding the maternal desires, the desire for the infant’s development (“You have to grow”) is only expressed for the daughter (so there is probably a greater expectation of her development) while the desire for hospital discharge (“We’re about to go home, aren’t we?”) and family union (“You will be with your mother, with your sister and with your grandparents, ok?”) are only expressed in speech addressed to the son (probably as a need for family support to care for the infant who is perceived as being more vulnerable).

Although these results are exploratory, they point to a maternal representation of higher levels of vulnerability and greater need for protection in relation to the son and a higher expectation of autonomy in relation to the daughter. As in previous studies, we found a greater tendency for mothers to support the self-regulation of male infants than of female infants [36,37]. More studies are needed to deepen the analysis of the verbal content of motherese directed to twin preterm infants.

## 5. Limitations

One of the main limitations of this exploratory study is the fact that it is a case study of only one mother–twin preterm infants’ dyad. In particular, this case study was of premature twins and of a single mother of an immigrant and professional background. These psychosocial variables represent a range of resilience and vulnerability factors for mother–infant interactions and the study is unable to isolate the impact of these variables. Maternal mental health is well established as affecting mother–infant interaction, which was not studied here.

A more robust study would be to compare this case study of twins with two singleton preterm infants, one male and one female.

Another limitation was the acoustic quality of the audio recording due to environmental noises (NICU’s monitors, staff interactions, etc.) which restricted the acoustic analysis of the female infant’s vocalizations.

## 6. Conclusions

The main contribution of this exploratory study was to increase knowledge about the multimodal interaction of a pair of premature twins. A key conclusion that can be drawn is that the mother interacts differently to the two preterm infants and that is influenced in part how she responses to infant gaze and infant’s vocalization. More studies are needed to understand the role of the infant’s gender, the infant’s behavior, as well as the individual differences of each infant to tonal modulation (increasing or decreasing pitch) of maternal speech and humming. The findings of our study may contribute to the general future parenting interventions for preterm infants particularly during the practice of Kangaroo care, and future research directions with respect to multimodal interaction as promising leads to optimize preterm infant outcomes. Given the exploratory nature of the study, one would have thought that the value in such a study is generating promising new leads for future studies in larger samples.

## Figures and Tables

**Table 1 children-08-00754-t001:** Descriptive analysis of the lengths (ms) of maternal vocalizations (phrases) and of touch units in speech or humming to twin 1 and to twin 2.

Vocal Condition	Maternal Behavior	*n*	Min.	Max.	M	SD
	to twin 1					
speech	vocalizations	72	200	4560	1514.09	919.27
	touch units	47	55.00	2610	1252.87	601.30
humming	vocalizations	72	790	4510	1880.86	840.64
	touch units	43	350	4950	2174.65	1113.58
	to twin 2					
speech	vocalizations	64	280	3871	1438.14	835.58
	touch units	3	910	1550	1310.00	348.71
humming	vocalizations	61	2000	2741	2355.50	179.94
	touch units	0	0	0	0	0

**Table 2 children-08-00754-t002:** Descriptive analysis of the lengths (ms) of infants’ behavior (vocalizations and gaze episodes) in speech and in humming.

Vocal Conditions	Infant Behavior	*n*	Min.	Max.	M	SD
	twin 1					
speech	gaze episodes	11	380	2820	1705.45	934.19
	vocalizations	9	130	4200	245.55	102.84
humming	gaze episodes	0	0	0	0	0
	vocalizations	28	100	5200	263.75	113.84
	twin 2					
speech	gaze episodes	4	770	4110	2737.50	1600.02
	vocalizations	0	0	0	0	0
humming	gaze episodes	5	890	2310	1726.00	562.78
	vocalizations	0	0	0	0	0

**Table 3 children-08-00754-t003:** Descriptive analysis of maternal pitch (Hz), in speech and in humming, before and after infant’s behavior.

Conditions		Maternal Pitch (Hz)			
		to twin 1			
		Min.	Max.	M	SD
speech	before infant gaze	202.56	242.84	223.01	16.11
	after infant gaze	206.88	269.59	230.31	25.34
	before infant vocalizations	228.67	268.05	250.96	20.19
	after infant vocalizations	189.86	261.70	224.66	35.97
humming	before infant gaze	0	0	0	0
	after infant gaze	0	0	0	0
	before infant vocalizations	232.01	299.71	249.28	28.78
	after infant vocalizations	249.16	267.03	255.06	7.41
		to twin 2			
		min.	max.	M	SD
speech	before infant gaze	190.67	224.66	209.87	15.93
	after infant gaze	209.26	251.42	233.59	19.67
	before infant vocalizations	0	0	0	0
	after infant vocalizations	0	0	0	0
humming	before infant gaze	146.93	240.35	201.88	40.82
	after infant gaze	50.65	241.09	156.44	106.79
	before infant vocalizations	0	0	0	0
	after infant vocalizations	0	0	0	0

**Table 4 children-08-00754-t004:** Descriptive analysis of melodic contour lengths (ms) in humming to twin 1 and to twin 2.

Melodic Contours	*n*	M	SD	min.	Max.
to twin 1 (*n* = 72)					
linear	0	0	0	0	0
rising	27	1112.11	485.87	790.00	2170.00
falling	0	0	0	0	0
bell-shaped	15	2190.00	282.08	1930.00	2900.00
U-shaped	6	2283.33	215.83	2020.00	2610.00
sinusoidal	25	2519.20	931.48	1880.00	4660.00
to twin 2 (*n* = 61)					
linear	0	0	0	0	0
rising	0	0	0	0	0
falling	0	0	0	0	0
bell-shaped	10	2359.20	102.07	2277.00	2574.00
U-shaped	11	2238.63	174.64	2055.00	2630.00
sinusoidal	39	2396.56	177.82	2111.00	2741.00

**Table 5 children-08-00754-t005:** Categories and subcategories of speech contents directed to twin 1 and to twin 2.

Categories of Speech	Subcategories of Speech	to twin 1	to twin 2
needs	infant’s needs	Hi! How was your night?	How are you there? Are you comfortable?
	maternal needs	Mother missed you.	Oh, my love don’t sleep! Mom came to see you!
Infant designation	first name	Where is Beatriz?	Hi David!
	diminutive	Bia	Not identified
	affectionate word	So my little princess?	Hello my Prince! Mom’s youngest!
physical features of infant	positive features	You’re opening your little eye, are you?	Not identified
	negative features	Not identified	Not identified
temperamental features of infant	positive features	Not identified	Not identified
	negative features	Very tired? Poor girl!	Not identified
maternal worries	infant’s sleep	Did you sleep well tonight?	Hmm! how sleepy I am!
	breastfeeding/weight	Did you eat a lot today?	Not identified
	comfort/temperature	Have you been feeling very cold?	Are you comfortable and warm in here?
	lap	My princess is on mommy’s lap	My youngest, are you enjoying lap of mom?
maternal desires	infant development	You have to grow	Not identified
	hospital discharge	Not identified	We’re about to go home, aren’t we?
	family union	Not identified	You will be with the mother, with the sister and with the grandparents, ok?

**Table 6 children-08-00754-t006:** Frequencies and proportions (according to the total of maternal speech vocalizations) of the speech contents directed to twin 1 and to twin 2.

Categories of Speech	Subcategories of Speech	to Twin 1 (*n* = 72)		to Twin 2 (*n* = 64)	
		Frequency	%	Frequency	%
needs	total	36	50	34	53.12
	infant’s needs	18	25	20	31.25
	maternal needs	18	25	14	21.88
infant designation	total	28	38.89	17	26.56
	first name	10	13.89	7	10.94
	diminutive	7	9.72	0	0
	affectionate word	11	15.28	10	15.63
physical features of infant	total	2	2.78	0	0
	positive features	2	2.78	0	0
	negative features	0	0	0	0
temperamental features of infant	total	1	1.39	0	0
	positive features	0	0	0	0
	negative features	1	1.39	0	0
maternal worries	total	11	15.28	18	28.13
	infant’s sleep	5	6.94	11	17.19
	breastfeeding/weight	1	1.39	0	0
	comfort/temperature	3	4.17	5	7.81
	lap	2	2.78	3	4.68
maternal desires	total	1	1.39	3	4.69
	infant development	1	1.39	0	0
	hospital discharge	0	0	2	3.13
	family union	0	0	1	1.56

## Data Availability

For information about the data contact the first author of the study (eduardacarvalho@fcsh.unl.pt).

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
