# Peer review of "Multimodal Interaction between a Mother and Her Twin Preterm Infants (Male and Female) in Maternal Speech and Humming during Kangaroo Care: A Microanalytical Case Study"

_children, 2021, doi:10.3390/children8090754_

Round 1

Reviewer 1 Report

Overall impression:

            This is a very interesting article asking multiple research questions that aid in the understanding of parent/infant interactions during neonatal care. There are differences in the interaction style between the male and female infant which is an important addition to the literature that has yet to be consistently studied or integrated into care practices.

Text/table revisions:

  • Method: it would be helpful for both context and future replication if the “kangaroo care with visual contact” was further described. How was the infant positioned on the mother’s body?
  • Sample: it is reported that the infants were born at 32 weeks gestation but it is not reported what age the infants were at the time of the study. Knowing the GA of the infants at the time of the study is important in the interpretation of the results as a 32 week infant has different capacities for interaction compared to an older infant. Please add this information to the sample.
  • Line 292: I think the 4 in (4, 68%) needs to be a 3 if it represents the n for the male infant that is listed in table 1.
  • Table 4: fix the alignment of “to female infant (n=72)” so it lines up with “to male infant”
  • Results: an additional table that provides qualitative examples of the actual words or phrases that were spoken to the infants would be helpful in more fully understanding this result

Discussion:

  • Lines 392-399: Segments vs. length of humming or speech – this is a summary of the results but it does not offer any additional discussion of this finding. Duration (length) here seems to be the most important concept and the mother hummed longer to the male infant and there was no difference in total duration for speech. This might have implications for auditory development (more time hearing mothers voice). Please tie back to your background literature on auditory development.
  • Lines 473-486 – can you point back to some of the actual phrases or words spoken to each infant?

Author Response

Please find below a response to each comment in the letter.

Reviewer 2 Report

This was an interesting microanalytic case study of a set of dizygotic twins focused on maternal speech and humming, and infant gaze and vocalisations. There is definite potential merit in this kind of detailed fine-grained study that is typically very time-consuming to analyse. However, unfortunately, I also have four major issues with the manuscript in its current form. These really do need addressing, which may require additional coding work for inter-rater reliability and substantial rewriting especially of the discussion as well as how the study is framed in relation to infant sex/gender in the introduction/abstract and elsewhere.

1) No description is given on the microanalytic coding schemes used and whether they have previously been validated (particularly pertinent for the verbal categories of maternal speech content, which I imagine would need to be meticulously operationalised, but relevant to all the coding schemes/categories employed), and no report of inter-rater reliability (IRR) is provided (typically reporting kappa based on the independent blinded second coding of a randomly selected sub-sample of recordings to see how well they matched up to the primary coder’s codings). Replicability and validity of the data rest hugely on the use of robust measures and the quality of training to the coder/s. Firstly, having good/high IRR is essential; without which, one cannot evaluate the quality of the data. If one cannot reach high IRR on some measure/s (I hope you do), then they really cannot be analysed and therefore careful consideration is needed around omission or treatment of the data in a different way. Secondly, if the measures were developed by the authors, then this needs to be made clear in the method section, a paragraph outlining the development of the coding scheme/s is necessary, plus reference to any availability of the (for example) coding manual/s by either contacting the author, or by providing a summarised version in the supplementary appendices/materials if this is permitted by the journal. Depending on the level of validity of the measures or their stage in development, it may also be appropriate to acknowledge potential limitations with the use of new measures.

2) One major issue I had from both the abstract and the main manuscript is the use of language when attributing findings to gender which are likely to be construed by readers as a generalisation. This could in part be due to English not being a first language for the authors (For example, ‘to male than female infant’ needs to be ‘to the male than to the female infant’ to be clear that these refer to two specific infants), but it seems that conclusions are actually drawn accordingly; e.g. re infant gender and infant behaviour. Fundamentally, the use of infant sex to differentiate each infant can be very confusing and it needs to be much clearer that the description provided is not necessarily attributed to infant sex/gender. I would prefer a different identifier, such as ‘twin 1 (f)’ and ‘twin 2 (m)’ so that gender is not their only identifier. Pertinently and relatedly, many conclusions drawn are questionable. Longer maternal vocalisations in humming toward (the) male infant could be due to a variety of reasons (e.g. born later and smaller so may be perceived as more vulnerable; more maternal concern / anxiety about the boy being ‘less strong’ given societal expectations especially given the woman’s background from an LMIC and of high socioeconomic status herself) or because the boy showed calmer/more attentive behaviour, encouraging the mother to continue humming longer). More affection to the daughter (caressed face) could bue due to gender, but just as much could be due to the infant’s behaviour (e.g. if more agitated) or the mother being more comfortable if that interaction took place after the one with the boy. The discussion should necessarily cover wider consideration of what the findings could mean, especially given this specific context.

3) While some details are provided, the data collection section still needs more detail as it’s currently not entirely clear how the data is collected, the context in which it was done and how it was analysed. It is too sparse for replication currently. What are vocalisations in humming? I may be wrong, but I thought humming was a kind of vocalisation? Is the humming the condition? It could be clearer why you had selected a structured set-up (silence, speech to infant, silence…) and what exactly was the mother instructed? Was she instructed to hum even though she may not ordinarily hum? It sounds like mother was asked to use visual contact, talking and humming in an improvised way. Is that so? Might the instructions have affected maternal behaviour, especially without counterbalancing due to it being a case study (e.g. mother may be more comfortable in the second interaction)? Where did this all take place? How much time elapsed between the 2 Kangeroo care experiences and how much instruction and guidance was provided and by whom? I also could not find precise infant age?

4) Related to point 2 above but deserves consideration on its own is the conclusions drawn from the case study findings. A key conclusion that can be drawn is that the mother interacts differently to the two preterm infants and that is influenced in part by (how she responses to) infant gaze and infant vocalisations. The very specific context of this case needs to within part of your reasons for this case study rather than hidden in the method section and may offer strengths and limitations. This is a very specific case: An older-than-average first-time single mother of Angolan origin with high occupational status, no current partner, and concern about sickle cell anaemia. Artificial insemination after previous recent spontaneous abortion indicates a very highly motivated parent yet likely understandably high levels of parental anxiety, concern or stress at having preterm twins. There are clear health indicator and perceived temperamental differences that we know also impact on maternal and infant interactive behaviour. Generally, the wording needs to be moderated and conclusions more considerate in scope and tentatively drawn. Also, take great care to avoid drawing conclusions from non-significant results (line 452).

The following are more minor issues:

  1. It’s a little disorienting to mention two theories in the opening paragraph without context. The theory/theories should only be mentioned if they somehow help provide some framework for the study.
  2. Vocal interactions ‘which ensures an interactive synchronization of the dyad’ (33/34) – doesn’t sound quite right. Dyads vary in synchronisation so how is this ensured? Newborns aren’t synchronous; rather, mother elicit infant engagement and familiarity with mother’s intonational characteristics?
  3. Small improvements needed in writing throughout. E.g. Line 39: ‘Relative’ – right word? Line 409: ‘Also the literature underline, a higher mutual synchrony’ – not grammatically correct. 460: ‘As a hypothesis, we can think that’ – vague; are you suggesting your findings suggest this?
  4. More references to strengthen this assertion? (Currently only ref 4).
  5. The intro section is quite long (I would suggest some parts could be more concise), yet no mention of infant-directed speech or Motherese which is a large part of the speech intonation and perhaps other characteristics referred to.
  6. The interview data mentioned in the method would be interesting to include? If this manuscript doesn’t deal with that data, then why do you mention in the method? Be clearer about this.
  7. 429: I don’t feel it can be said that your findings don’t agree with another study that is concerned with a different age range. Singing by mothers is likely to be used for different reasons for older infants from younger infants. Dyadic play is reduced except in singing games/routines. Further, little is known about cultural differences, despite the universality of singing, and this should be acknowledged.
  8. A more robust study would be to compare this case study of twins with 2 singleton preterm infants, one male and one female. From this, we could be more confident about any conclusions drawn. This could be acknowledged in the discussion/limitations.
  9. The limitations could be much more fleshed out (mentioned in several earlier points). Line 493: male didn’t produce any at all – what does that mean?

I do not know of PRAAT, the acoustic analysis software, so cannot comment on this.

Thanks for the opportunity to review this paper. Essentially there is value in this paper potentially, but more work is needed to demonstrate robustness of the findings, replicability of methods and a much more balanced and deeply considered discussion.

Author Response

(The authors gave the same response as above.)

Round 2

Reviewer 2 Report

Thank you for addressing the comments I highlighted in my previous review. The major comments were, on the whole, addressed satisfactorily. The paper has been substantially strengthened as a result and I’m pleased to say that I have no more major comments. However, I do have some comments which, if addressed, could further substantially improve the rigor of the paper. Note especially that I felt the aims as they currently stand could do with more detail. Even though/if framed as an exploratory study, it could be a lot clearer in the rationale why the areas of communication/interaction were chosen and studied in the ways chosen. The other key points to highlight here are taking care with wording/interpretation in the discussion not to overstate the findings, enriching the limitations some more, and condensing the results section. You do repeat in the discussion that this is an exploratory study; however, the wording of the interpretation still needs to be cautious and the conclusions could benefit from more development to help polish the paper and for it to be truly informative.

My comments:

1) There are a few (small but cumulative) issues early on that could be rectified easily or relatively easily to make a better first impression. Integrate the second paragraph with the first paragraph to provide a more integrated and professional look to your paper. It would make more grammatical sense to phrase as “Mother-infant interactions are coordinated in a multi-modal way…” Also, briefly define what you mean by ‘multi-modal’. I see that you refer to this later on, but it needs some context earlier in the introduction where you first mention it. In the first sentence of paragaph 1, ‘infants’ would be more appropriate than ‘children’ as singing etc. drops off as children acquire language and speech.

2) Lines 35 and 126 – increases/great vulnerability to what? Also line 146 – clarification needed.

3) Line 98: “the parental use of affectionate stroking frequently” – Should this read “frequent affectionate stroking increases the infant’s…” Ambiguous.

4) Line 109 onward: Please check through the appropriateness of how previous study findings are reported re how mothers interact with their daughters vs sons. The suggestion made is that mothers are less synchronous with sons than daughters and sons take more time to orientate etc. during interaction, and this is an important part of the study rationale so important to get right. Reference 37, one key citation used by the authors to support that point, is a qualitative study of interviewing mothers about their experiences of a particular intervention when their child (aged from newborn to around 3.5 years) was undergoing invasive procedures in a pediatric intensive care unit, and thus was critically ill. Neither the methods nor the population is relevant to the statement made. Reference 38 (Feldman’s study) focused on mother- and father-infant interactions and found a greater time-lag-to-synchrony in opposite-sex interactions; so I agree that mother-son interactions had a larger lag time (vs mother-daughter interactions), but the current wording suggests that the difference in initiated by the sons, yet (1) the lag time was determined by one partner following the other who was leading, it was not defined as the mother following the son specifically, so could be due to mothers’ slower engagement to sons’ initiations; (2) father-son lag time was significantly shorter than father-daughter lag time, further suggesting that the lag time was nothing inherent in the boy infants’ social orientation capacities. The interpretative issues here need to be rectified because this affects the reader’s confidence in the other citations. Overall, this also raises a general question of whether observations are due to infant sex differences or differential parental responses to infants of different sexes (or gender socialisation).

5) I found the aims very broad and therefore did not really understand what your aims were. Please rephrase in a more conventional way. For example, the authors proposed: “we intend to know about: 1) the maternal vocalizations in humming and speech conditions” – what about them? Whether they exist? Whether they were longer or more of them in one condition than another? The same query applies for all the objectives stated. It may be helpful for the authors to really think about their focus here - is it on prematurity as a risk factor or about multimodal interaction? 

6) While it’s clear that research on multi-modal communication / cues during mother-infant interaction has not been widely researched in infants, it is unclear in your introduction why the focus on e.g. affectionate touch and maternal phrases in speech and in humming. In other words, more detail would be beneficial to understand why those areas of communication were chosen. Currently the introduction seems to focus more on synchrony which does not appear to be your focus.

7) Line 192+, “I will ask you to alternate periods where you will be silent with others periods where you will talk to your infant..” There seem to be multiple typos or language/phrasing issues here. “IN alternative periods WHEN” etc? Also, in line 196, how did counterbalancing take place?

8) Line 204: “In order to deepen the features of the multimodal interaction a video and audio recording using a camera” – I don’t understand what you mean by deepen the features. Line 206: I would call these sets of microanalyses or microanalytic codings, rather than microanalysis studies.

9) Table 1: Take care with grammar (e.g. “An infant”; capitalisation of first word in headings).

10) Thank you for the reliability information. Was the second coding completed for the full dataset or for a proportion? Please clarify. I note the high variability of df.

11) Results section: With 6 tables and a fair amount of text, and especially as this is a case study, I would suggest reducing the number of tables and perhaps condensing the text to help create more impact. Consider combining tables. Do we really need mins and maxes? I would not say that is the convention in published papers, especially of case studies.

12) I would like to see in the first paragraph of the discussion a summary of all the key findings in relation to the specific objectives set out.

13) Line 425: What do you mean by ‘segmentation’? Try to consider accessibility to a wider audience.

14) Line 428: I would prefer a phrasing like “The pattern of findings in this case study suggests that the mother may have perceived twin 2 as more vulnerable…” Generally, a cautious phrasing is preferred when the study design does not allow for the test of causal relationships. “It is probable” suggests a pretty high level of certainty that a case study cannot provide. Do you have supporting information from the interview/s? This issue applies throughout the discussion. I think it’s reasonable to say that there are some points made that may be viewed by the scientific community as overstated. For example: from Line 447+, “No significant differences were observed in the length of the speech vocalizations directed to twin 1 compared to the vocalizations directed to twin 2. This suggests that the use of humming seems to be more favorable than speech for promoting contingent interaction and tuning of a preterm dyad.” I do not understand how one can draw that conclusion from the lack of significant differences on something else. How do we know that maternal speech is less important for promoting contingent interaction than humming without directly comparing them?

15) In the discussion, care is needed to distinguish behavioural differences between the two twins which may be attributed to their gender/sex or/and other individual differences. While both are mentioned, it is worth being very clear about these different possible interpretations.

16) As noted previously, I do think the limitations section is very brief and some more considered reflection would only strengthen the study, so I would strongly encourage that. In particular, this case study was of premature twins and of a single mother of an immigrant and professional background. These psychosocial variables represent a range of resilience and vulnerability factors for mother-infant interactions and the study is unable to isolate the impact of these variables. Maternal mental health is well established as affecting mother-infant interaction, which was not studied here.

17) There are some typos or grammatical errors in the conclusions. Overall, I like the new conclusion. However, in both the intro and conclusion, there is reference to the need for programs for promoting multimodal interaction among mothers and premature twins. That seems extremely niche and with all the will in the world, it seems extremely unlikely that there will be programs developed or widely implemented for that very specific group. It may be more convincing to frame this in terms of how the findings may inform (1) general future parenting interventions for premature infants and perhaps the practice of Kangaroo care; and (2) future research directions with respect to multimodal interaction (which areas specifically?) as promising leads to optimise premature infant outcomes. Given the exploratory nature of the study, one would have thought that the value in such a study is generating promising new leads for future studies in larger samples and theory development.

Author Response

Reply to reviewer 2
Thank you for acknowledging the progress of this manuscript and for encouraging its improvement. The suggestions and comments were very helpful.
Follow the answers numbered according to each comment:
1) This point has been corrected (29-35).
2) The sentences have been changed (38-39, 147-148).
3) The sentence has been changed (96-98).
4) I didn't fully understand your comments, but I think there was a mistake in the numbering of the bibliographic references that perhaps would not have facilitated the understanding of the text.
5) The aims have been reformulated (129-141).
6) The justification regarding the analysis of maternal touch during humming and speech was added (123-128).
7) The instruction to the participants was reformulated in a more systematic way and the offset was explained in the order of the sequence (193-203).
8) The reference to the material (video camera, etc.) has been transferred to the Design sub section (190-191) and terminology “microanalysis studies” was changed for “microanalytic coding” included the subsection microanalytic coding (208-209).
9) Table 1 was deleted and its contents were transferred to the text (216-226).
10) All encodings were filled in by each of the two investigators for the complete dataset (df is the number of encodings for each variable and in the case of maternal vocalizations we actually found a higher number according to the established criterion of 300ms of pauses) and the reliability data (256-280) was only based on the case study sample and not on the larger sample.
11) The number of tables was reduced to 6 tables and table 1 was deleted. (minimum and maximum values can help to understand the durations of the variables).
12) Regarding the proposed aims of this study the most significant or relevant results were written in the first paragraph of the Discussion section (427-441).
13) The term “segmentation” was deleted for not contributing relevant information because when we say that the humming phrases are longer this implies that the phrases were less interrupted. In any case, the meaning of segmentation is well documented in the literature on temporal analysis of behaviors (Condon, WS; Ogston, WD; “A Segmentation of Behavior.” Journal of Psychiatric Research, 1967, 5, 3, 221–35. 601 https ://doi.org/10.1016/0022-3956(67)90004-0).
14) This point was rephrased according to your writing suggestion (437-441). Also non-significant results were deleted from the discussion and their interpretations were also deleted.
15) I agree that in addition to the contributions of the gender and behavior of the babies, there is a reference to the possible contribution of the individual differences of each baby and that is why this aspect of individual differences was added (478-480).
16) In fact, this study gave priority to the microanalysis of the observed behaviors of dyad, not considering the psychosocial variables associated with the mental health of the particular case study, thus adding to the limitations of the study (514-519).
17) . The conclusion was reformulated and the most relevant aspects about the contribution of the study were synthesized according to its writing suggestions (526-537).
